# Some Physical Properties and Mass Modelling of Pepper Berries (*Piper nigrum* L.), Variety Kuching, at Different Maturity Levels

**Puteri Nurain Megat Ahmad Azman [1], Rosnah Shamsudin [1,2,\*], Hasfalina Che Man [3] and Mohammad Effendy Ya'acob [1]**

1   Department of Process and Food Engineering, Faculty of Engineering, Universiti Putra Malaysia, Serdang 43400, Malaysia; gs53584@student.upm.edu.my (P.N.M.A.A.); m_effendy@upm.edu.my (M.E.Y.)
2   Institute of Advanced Technology, Universiti Putra Malaysia, Serdang 43400, Malaysia
3   Department of Biological and Agricultural Engineering, Faculty of Engineering, Universiti Putra Malaysia, Serdang 43400, Malaysia; hasfalina@upm.edu.my
\*   Correspondence: rosnahs@upm.edu.my; Tel.: +60-0397696366

**Abstract:** Pepper berry (*Piper nigrum* L.) is known as the king of spices and has sharp, pungent flavour and aroma. In this study, the physical properties (weight, dimensions, sphericity, volume, surface area, and projected area) were measured, and the mass of pepper berries of the Kuching variety at different maturity levels (immature, mature, and ripe) was predicted using four models: linear, quadratic, s-curve, and power. When the models were based on volume and projected area, the mass could be predicted with maximum precision. The Quadratic model was best fitted for mass prediction at all mass maturity levels (immature, mature, and ripe). The results showed that mass modelling based on the actual volume of pepper berries was more applicable compared to other properties with the highest determination coefficient, 0.995, at the 1% probability level. From an economical point of view, mass prediction based on actual volume in the Quadratic form, $M = 0.828 - 0.015\,V + 7.376 \times 10^{-5}V^2$, is recommended. The findings of physical properties and mass modelling of the berries would be useful to the scientific knowledge base, which may help in developing grading, handling, and packaging systems.

**Keywords:** piper nigrum; dimensions; mass; maturity levels; modelling

## 1. Introduction

Pepper (*Piper nigrum* L.) is a perennial climber belonging to the family Piperaceae. The leafy pepper tree is comparable to an almond tree that is able to grow tall, reaching a height of 10 m or more. When the main stem matures, many side shoots start growing on it to make a dense canopy. Pepper has a sharp, pungent aroma and flavour. It is widely planted in Vietnam, Malaysia, Indonesia, Sri Lanka, and other areas. According to the Food and Agriculture Organisation of the United Nations Statistics [1], Malaysia was the world's seventh largest pepper producer in 2017. About 98% of pepper production comes from Sarawak, the largest national producer; the other 2% is produced in Johor and other states in Malaysia.

A few recommended local pepper varieties include Kuching, Semongok Emas, and Semongok Aman [2]. Kuching pepper is the most commonly grown cultivar in Sarawak and Johor, Malaysia, due to its high and stable yield. It also has a denser canopy and thinner pericarp, which is the preferred variety for white pepper, production as compared to others. Semongok Emas and Semongok Aman are often used for black pepper production due to their pungent flavour, high yield, and thicker pericarp. In the harvesting process, the selection of pepper berries depends on the maturity levels such as immature, mature, and ripe. Immature pepper berries are described as light green. Mature

pepper berries are dark green and yellowish-green, and ripe pepper berries are red. Black pepper and white pepper are the most common and well-known worldwide. For producing black pepper, dark green pepper berries (mature pepper berries) with soft peppercorns and sharp, pungent aroma and flavour are used. They will undergo the drying process without removing the outer skin (pericarp). Yellowish-green and red pepper berries (mature and ripe pepper berries) are used for white pepper production due to the hardness of the peppercorn and thinner pericarp. They are produced by removing the outer skin (pericarp) after a soaking process, and the peppercorns will undergo a drying process. In general, the average diameter of mature and ripe pepper berries for all varieties is about 5 to 6 mm. Overall, the size of pepper berries is similar at all maturity levels. Furthermore, pepper, referred to as black pepper and white pepper, is used in Ayurvedic remedies for health benefits (appetite stimulant, breathing aid, cough therapy, anaemia, and others), keeping food fresh by combining with salt to cure and flavour a wide variety of meats (early food-preservation techniques), and reducing pain perception and inflammation. It is also rich in vitamins and minerals.

Although the berries are often graded by size, weight-based grading is essential in packaging and handling because it may be more economical. Therefore, the physical properties and the relationships must be determined to design and optimise a machine for handling, cleaning, conveying, and storing [3]. The most crucial properties in the design of the grading system are dimension, mass, volume, and projected area [4,5]. According to Pradhan et al. [6] and Pathak et al. [7], basic data on the physical properties of bio-materials are useful to all engineers, industrial processors, scientists, pharmacists, crop breeders, researchers, and others who may discover their advanced usage. Analysis of regression produces an equation to describe and predict the statistical relationship between one or more predictor factors and the variable of reaction. Some regression relationships, such as Linear, Quadratic, S-curve, and Power, were mostly used in previous studies. The statistical significance in the regression relationship shows that the dependent variable modifications in the dependent variables correlate with shifts. Therefore, a high coefficient of determination ($R^2$) indicates that the model explains a decent percentage of the variability in the dependent variable. The correlation determination technique between mass and its physical properties is more specific for automatic classification of most berries and fruits.

There is a lack of previous research on mass prediction by using model equations for berries compared to other fruits. A few differences between berries and other fruits can be observed. The berries are fleshy fruits produced from a single ovary, while other fruits can be produced from single or multiple ovaries. The berries are edible when the entire ovary wall is ripened and yet, it may not be the same for all fruits. The study of mass modelling of pepper berries has not been carried out in detail in published works. Thus, this research aims to determine some physical properties such as the dimensions (major, medium and minor axis), volume, aspect ratio, geometric mean diameter, sphericity, surface area, and projected areas of immature, mature, and ripe pepper berries. It also serves to determine the most applicable model for predicting the mass of pepper berries at different maturity levels to form an essential database for other researchers. To the authors' knowledge, two previous published works concerned the mass modelling of berries based on their physicochemical properties such as raisin berries [8] and sea buckthorn berries [9]. In other research, many valuable studies were concerned about the mass modelling of fruits based on their physicochemical properties that included oranges [10], apples [11], pomegranates [12], onions [13], fava beans [3], persimmons [5], pomelos [14], dates [15], Kinnow mandarin [16], chebula fruits [7], and banana [17]. In summary, the study findings may help in developing grading, handling, and packaging systems based on the physical properties at different maturity levels of pepper berries.

## 2. Materials and Methods

The Kuching variety of pepper berries was obtained from a farm located in Johor, south of Malaysia (Johor, Malaysia). The fresh pepper berries were immediately transported to the laboratory within 4 h. The pepper berries were selected and divided into three maturity levels: immature, mature, and ripe. For sample selection, colour is an essential visual parameter to differentiate pepper based on the maturity levels. Light green and red berries were labelled as immature and ripe pepper berries, respectively [2]. Dark green and yellowish-green berries were labelled as mature pepper berries [2]. One hundred samples of pepper berries from each maturity level were selected and used for physical property measurements [18,19] as shown in Table 1. All the experiments were conducted at room temperature in the laboratory.

### 2.1. Measurements of Physical Properties

Pepper berry mass ($M$) was determined with 0.01 g sensitivity of an electronic balance. Three linear dimensions, namely, the major axis ($L$), medium axis ($T$), and minor axis ($W$), were measured by using a digital vernier calliper with 0.01 mm sensitivity to determine the average size of the samples. The method of water displacement [1,3,12,20,21] was used to determine the actual volume ($V$). The uniform presumed shape of the fruit can be an aspect to determine the fruit volume accuracy [5]. The aspect ratio ($AR$), geometric mean diameter ($D_g$), and sphericity ($\Phi$) were calculated by using the following respective formulas [5,20–22]:

$$AR = \frac{L}{W} \tag{1}$$

$$D_g = (LWT)^{\frac{1}{3}} \tag{2}$$

$$\phi = \frac{(LWT)^{\frac{1}{3}}}{L} \tag{3}$$

The surface area is defined as the total three-dimensional shape areas of all surfaces. Equation (4) was used to calculate the spheroid surface area of pepper berries.

$$SA_{sp} = 4\pi r^2 \tag{4}$$

The projected areas of pepper berries ($PA_L$, $PA_T$, and $PA_W$) in three perpendicular directions to the dimensions (major axis, medium axis, and minor axis) and the criteria projected area ($CPA$) were calculated by using Equations (5)–(8), respectively. These equations were suggested by Mohsenin [4] and Nur Salihah et al. [14] and are defined as follows:

$$PA_L = \frac{\pi LW}{4} \tag{5}$$

$$PA_T = \frac{\pi TW}{4} \tag{6}$$

$$PA_W = \frac{\pi WW}{4} \tag{7}$$

$$CPA = \frac{PA_L + PA_T + PA_W}{3} \tag{8}$$

**Table 1.** Some physical properties of pepper berries at different maturity levels.

| Parameter | Immature | | | Mature | | | Ripe | | |
|---|---|---|---|---|---|---|---|---|---|
| | Mean | Maximum Value | Minimum Value | Mean | Maximum Value | Minimum Value | Mean | Maximum Value | Minimum Value |
| $L$ (mm) | 4.75 ± 0.38 [c] | 5.50 | 4.00 | **5.73 ± 0.32** [a] | 6.60 | 5.40 | 5.55 ± 0.66 [b] | 7.40 | 4.36 |
| $T$ (mm) | 5.09 ± 0.50 [c] | 5.70 | 4.30 | **6.16 ± 0.44** [a] | 7.10 | 5.60 | 5.60 ± 0.33 [b] | 7.10 | 4.31 |
| $W$ (mm) | 4.74 ± 0.34 [c] | 5.30 | 4.00 | **5.87 ± 0.34** [a] | 6.70 | 5.40 | 5.67 ± 0.51 [b] | 7.01 | 4.36 |
| $AR$ | **1.00 ± 0.03** [a] | 1.04 | 0.96 | 0.98 ± 0.02 [a] | 1.00 | 0.93 | 0.98 ± 0.13 [a] | 1.18 | 0.79 |
| $D_g$ (mm) | 4.85 ± 0.38 [c] | 5.50 | 4.10 | **5.92 ± 0.35** [a] | 6.80 | 5.47 | 5.61 ± 0.46 [b] | 7.17 | 4.67 |
| $\Phi$ | 1.02 ± 0.02 [ab] | 1.05 | 0.97 | **1.03 ± 0.02** [a] | 1.08 | 1.01 | 1.01 ± 0.08 [b] | 1.43 | 0.72 |
| Weight (g) | 0.10 ± 0.01 [a] | 0.11 | 0.09 | 0.14 ± 0.01 [a] | 0.14 | 0.13 | **0.15 ± 0.04** [a] | 0.23 | 0.11 |
| $V$ (mm$^3$) | 96.67 ± 5.77 [b] | 100.00 | 90.00 | **120 ± 10.00** [a] | 130.00 | 110.00 | **120 ± 21.60** [a] | 140.00 | 90.00 |
| $SA_{sp}$ (mm$^2$) | 74.65 ± 11.43 [c] | 95.03 | 52.81 | **110.49 ± 13.66** [a] | 145.27 | 93.88 | 99.39 ± 16.66 [b] | 161.51 | 69.20 |
| $PA_L$ (mm$^2$) | 17.78 ± 2.75 [c] | 22.89 | 12.57 | **26.50 ± 3.28** [a] | 34.73 | 22.90 | 24.78 ± 4.27 [b] | 40.74 | 17.70 |
| $PA_T$ (mm$^2$) | 19.04 ± 3.08 [c] | 23.73 | 13.51 | **28.51 ± 3.93** [a] | 37.36 | 23.75 | 25.10 ± 4.51 [b] | 39.09 | 14.96 |
| $PA_W$ (mm$^2$) | 17.73 ± 2.57 [c] | 22.06 | 12.57 | **27.16 ± 3.45** [a] | 35.26 | 22.90 | 25.45 ± 4.56 [b] | 38.59 | 14.93 |
| $CPA$ (mm$^2$) | 18.19 ± 2.73 [c] | 22.89 | 12.88 | **27.39 ± 3.52** [a] | 35.78 | 23.18 | 25.11 ± 4.13 [b] | 39.48 | 16.29 |

Data are expressed as the mean ± SD maximum minimum; $L$, major axis; $T$, medium axis; $W$, minor axis; $AR$, aspect ratio; $D_g$, geometric diameter; $\Phi$, sphericity; weight; $V$, actual volume; $SA_{sp}$, spherical surface area; $PA_L$, projected area perpendicular to major axis; $PA_T$, projected area perpendicular to medium axis; $PA_W$, projected area perpendicular to minor axis; $CPA$, criteria projected area. Different letters in a row indicate statistically significant differences at $p < 0.001$. Means that do not share a letter are significantly different. Tukey's test was applied with 95% confidence intervals.

## 2.2. Regression Analysis and Mass Modelling

In order to predict the mass of pepper berries based on measured physical properties (dimensional characteristics, volume, surface area, and projected area), the considerations of model classifications are as follows:

1.  Single variable regression of pepper berry mass based on dimensional characteristics of the pepper berry—major axis ($L$), medium axis ($T$), minor axis ($W$), and geometric mean diameter ($D_g$).
2.  Single regression of pepper berry volume—actual volume ($V$).
3.  Single regression of pepper berry surface area—surface area of the fruit assumed as a spheroid ($SA_{sp}$).
4.  Single variable regression of pepper berry projected area —$PA_L$, $PA_T$, $PA_W$, and $CPA$.

Four models, namely, Linear, Quadratic, S-curve, and Power, were used and fitted with the results obtained from the experiments. These models are presented in Equations (9)–(12), respectively [3,5]:

$$M = a + bX \tag{9}$$

$$M = a + bX + cX^2 \tag{10}$$

$$M = a + \frac{b}{X} \tag{11}$$

$$M = aX^b \tag{12}$$

where $M$ = *mass* (g); $X$ = the value of an independent parameter (physical properties) to find the relationship of it with mass; $a$, $b$, and $c$ = curve fitting parameters which are different in each equation.

## 2.3. Statistical Analysis

Data analysis and mass modelling prediction were performed by using statistical software such as SigmaPlot (Version 12.0) with significance at the 1% probability level. Coefficient of determination ($R^2$) and standard error of the estimate ($SEE$) were selected as the criteria to evaluate the applicability of the regression models. The applicable models were selected as those with higher $R^2$ and lower $SEE$ values [21].

## 3. Results and Discussion

### 3.1. Physical Properties of Pepper Berries

Table 1 summarizes the results of the physical properties of pepper berries based on the maturity levels. The mean values of the major axis ($L$), medium axis ($T$), and minor axis ($W$) for immature pepper berries were 4.75 mm ± 0.38, 5.09 mm ± 0.50, and 4.74 mm ± 0.34, respectively. Therefore, the medium axis had the highest mean value as compared to the major and minor axes of immature pepper berries. The mature pepper berries had mean values of 5.73 mm ± 0.32, 6.16 mm ± 0.44, and 5.87 mm ± 0.34 for major axis, medium axis, and minor axis, respectively. Again, the medium axis had the highest mean value as compared to the major and minor axes of mature pepper berries. Furthermore, the mean values of the major axis, medium axis, and minor axis for ripe pepper berries were 5.55 mm ± 0.66, 5.60 mm ± 0.58, and 5.67 mm ± 0.51, respectively. For the results of dimensional characteristics, the minor axis had the highest mean value as compared to the major and medium axis of ripe pepper berries. As can be seen, mature pepper berries had the highest values of the major, medium, and minor axes among these three maturity levels.

The mean value of the aspect ratio ($AR$) of immature pepper berries was 1.00 with a standard deviation of 0.03. The mature pepper berries had a mean value of 0.98 with a standard deviation of 0.02 for the aspect ratio. The mean value of the aspect ratio of ripe pepper berries was 0.98 with a standard deviation of 0.13. However, the value of the aspect ratio for immature, mature, and ripe pepper berries was significantly in same range as shown in Table 1.

The mean value of the geometric mean diameter ($D_g$) of immature pepper berries was 4.85 mm with a standard deviation of 0.38. Mature pepper berries had a mean value of 5.92 mm with a standard deviation of 0.35 for the geometric mean diameter. Furthermore, the mean value of the geometric mean diameter of ripe pepper berries was 5.61 mm with a standard deviation of 0.46. Thus, the mean value of the geometric mean diameter of mature pepper berries was the highest among all three maturity levels.

Sphericity can be described a solid shape formed relative to that of the same volume of a sphere. The ideal shape of the sphere is a solid with a high sphericity value [4,18]. A sphericity value of 1 is an ideal sphere. The mean sphericity value for immature pepper berries was 1.02 with a standard deviation of 0.02. The mature pepper berries had a mean value of 1.03 with a standard deviation of 0.02 for sphericity. As for ripe pepper berries, the mean value of sphericity was 1.01 with a standard deviation of 0.08. Thus, from the mean values of sphericity for all three maturity levels, it can be considered that the shapes of immature, mature, and ripe pepper berries were an ideal sphere. The mean weight of immature pepper berries was 0.10 ± 0.01 g. The mature pepper berries had a mean weight of 0.14 ± 0.01 g. The mean weight of ripe pepper berries was 0.15 g ± 0.04 g. However, the weight of immature, mature, and ripe pepper berries was significantly in the same range as shown in Table 1.

Based on Table 1, other physical properties of pepper berries included actual volume. The immature pepper berries had mean actual volumes of 96.67 mm$^3$ ± 5.77. The mean actual volume of mature pepper berries was 120 mm$^3$ ± 10.00. Furthermore, the ripe mature pepper berries had mean values of 120 mm$^3$ ± 21.60 for the actual volume. Overall, the mature and ripe pepper berries had the highest mean actual volumes when compared to the immature pepper berries.

The mean value of the spheroid surface area of immature pepper berries was 74.65 mm$^2$ ± 11.43. The mature pepper berries had a mean value of 110.49 mm$^2$ ± 13.66 for the spheroid surface area. The mean value of the spheroid surface area of ripe pepper berries was 99.39 mm$^2$ ± 16.66. Thus, the highest mean value of the surface area for the spheroid was obtained for mature pepper berries as shown in Table 1.

Table 1 shows the results of mean projected areas, perpendicular to the major axis ($PA_L$), medium axis ($PA_T$), and minor axis ($PA_W$). The results obtained for immature pepper berries were 17.78 mm$^2$ ± 2.75 ($PA_L$), 19.04 mm$^2$ ± 3.08 ($PA_T$), and 17.73 mm$^2$ ± 2.57 ($PA_W$). The mean values of $PA_L$, $PA_T$, and $PA_W$ for mature pepper berries were 26.50 mm$^2$ ± 3.28, 28.51 mm$^2$ ± 3.93, and 27.16 mm$^2$ ± 3.45, respectively. As for the ripe pepper berries, the mean values were 24.78 mm$^2$ ± 4.27, 25.10 mm$^2$ ± 4.51, and 25.45 mm$^2$ ± 4.56 for $PA_L$, $PA_T$, and $PA_W$, respectively. Therefore, the mature pepper berries had the highest mean values of $PA_L$, $PA_T$, and $PA_W$ when compared to the other two maturity levels of pepper berries. The criteria of projected area were determined by using the results of $PA_L$, $PA_T$, and $PA_W$ as indicated in Equation (11). The mean values of projected area criteria for immature, mature, and ripe pepper berries were 18.19 mm$^2$ ± 2.73, 27.39 mm$^2$ ± 3.52, and 25.11 mm$^2$ ± 4.13, respectively. Thus, the projected area criteria for mature pepper berries had the highest mean value in Table 1.

Projected area values are fundamental information in the design and development of machine vision-based grading systems [16]. Projected area is also useful for estimating the respiration rate, maturity index, and gas permeability to predict optimum harvest time, water loss, and heat and mass transfer during drying and cooling [7,16]. However, some ellipsoidal shapes of ripe pepper berries were observed among the samples. Also, the measured distinction in the physical properties may be due to the inherent variation in fruit dimensional characteristics.

*3.2. Mass Modelling*

The average dimensions, volumes, weight, surface areas, and projected areas of pepper berries obtained were used in mass modelling. Tables 2–4 show the best models obtained and their coefficients of determination, $R^2$, and *SEE* to predict mass by using the measured average dimensions, volumes, weight, surface areas, and projected areas of pepper berries. The correlations of physical properties with pepper berry mass as shown from the results obtained were significant at the 0.01 probability level. The regression mass was evaluated by using the coefficient of determination ($R^2$), where the best fit model was shown with a higher $R^2$ value (near 1.00).

*3.3. Models Based on Dimensions*

According to Table 2, the major axis ($L$), medium axis ($T$), minor axis ($W$), and geometric mean diameter ($D_g$) showed that the Quadratic model was the best-fit model to calculate and evaluate the mass of immature, mature, and ripe pepper berries. Table 2 shows the fitted models based on dimensions such as $L$, $T$, $W$, and $D_g$ with the values of $R^2$ and *SEE*.

For immature pepper berries, $D_g$ had a highest value of $R^2$ and the lowest value of *SEE,* which were 0.938 and 0.002, respectively, as indicated in Table 2. Equation (13) shows the equation of the Quadratic model obtained.

$$M = 0.225 - 0.071\, D_g + 0.009\, D_g{}^2 \tag{13}$$

According to Table 2, the mature pepper berries had $W$ and $D_g$ values with the highest $R^2$ (0.960) and the lowest *SEE* (0.001). The model equation obtained for these parameters was Quadratic. Equations (14) and (15) were determined for the parameters $W$ and $D_g$.

$$M = 0.260 - 0.053\, W + 0.005\, W^2 \tag{14}$$

$$M = 0.196 - 0.032\, D_g + 0.004\, D_g{}^2 \tag{15}$$

$L$ had the highest $R^2$ (0.980) and the lowest *SEE* (0.007) values for ripe pepper berries when compared to the others as given in the Quadratic model equation (Equation (16)).

$$M = 0.351 - 0.107\, L + 0.012\, L^2 \tag{16}$$

A similar model for onion and a Malaysian variety of pomelo fruit in another study was suggested and reported by Ghabel et al. [13] and Nur Salihah et al. [14], where the best model for mass determination based on $L$ was a Quadratic model following Equations (17) and (18):

$$M = 36.137 - 1.64\, L + 0.35\, L^2,\ R^2 = 0.96 \tag{17}$$

$$M = 28500.33 - 417.191\, L + 1.587\, L^2,\ R^2 = 0.993 \tag{18}$$

For the entire dimensions, the S-curve model was reported to have lower $R^2$ values as compared to other fitted models. The lower $R^2$ values could be due to the non-uniform mass of pepper berries corresponding to their size. Thus, the sizing of pepper berries based on length is recommended.

Table 2. Mass models of pepper berries based on dimensions.

| Dependent Parameter | Independent Parameter | Model Equation | Maturity Levels | Regression Constant | | | Statistical Parameters | | The Best Fitted Model |
|---|---|---|---|---|---|---|---|---|---|
| | | | | a | b | c | $R^2$ | *SEE* | |
| $M$ (g) | $L$ (mm) | Linear | Immature | 0.021 | 0.016 | - | 0.828 | 0.003 | Quadratic |
| | | | Mature | 0.058 | 0.013 | - | **0.949** | **0.001** | |
| | | | Ripe | −0.056 | 0.036 | - | 0.894 | 0.014 | |
| $M$ (g) | $L$ (mm) | Quadratic | Immature | 0.130 | −0.031 | 0.005 | 0.852 | 0.003 | |
| | | | Mature | −0.013 | 0.037 | −0.002 | 0.952 | 0.001 | |
| | | | Ripe | 0.351 | −0.107 | 0.012 | **0.980** | **0.007** | |
| $M$ (g) | $L$ (mm) | S-curve | Immature | 0.168 | −0.338 | - | 0.781 | 0.004 | |
| | | | Mature | 0.218 | −0.478 | - | **0.951** | **0.001** | |
| | | | Ripe | 0.345 | −1.085 | - | 0.796 | 0.020 | |
| $M$ (g) | $L$ (mm) | Power | Immature | 0.028 | 0.786 | - | 0.824 | 0.003 | |
| | | | Mature | 0.049 | 0.580 | - | **0.950** | **0.001** | |
| | | | Ripe | 0.012 | 1.432 | - | 0.911 | 0.013 | |
| $M$ (g) | $T$ (mm) | Linear | Immature | 0.033 | 0.012 | - | 0.862 | 0.003 | Quadratic |
| | | | Mature | 0.076 | 0.009 | - | **0.903** | **0.002** | |
| | | | Ripe | −0.052 | 0.035 | - | 0.795 | 0.020 | |
| $M$ (g) | $T$ (mm) | Quadratic | Immature | 0.239 | −0.071 | 0.008 | 0.925 | 0.002 | |
| | | | Mature | 0.286 | −0.057 | 0.005 | **0.952** | **0.001** | |
| | | | Ripe | 0.419 | −0.133 | 0.015 | 0.883 | 0.017 | |
| $M$ (g) | $T$ (mm) | S-curve | Immature | 0.155 | −0.296 | - | 0.821 | 0.003 | |
| | | | Mature | 0.194 | −0.370 | - | **0.871** | **0.002** | |
| | | | Ripe | 0.334 | −1.032 | - | 0.711 | 0.024 | |
| $M$ (g) | $T$ (mm) | Power | Immature | 0.033 | 0.653 | - | 0.856 | 0.003 | |
| | | | Mature | 0.060 | 0.439 | - | **0.895** | **0.002** | |
| | | | Ripe | 0.012 | 1.419 | - | 0.809 | 0.019 | |

**Table 2.** *Cont.*

| Dependent Parameter | Independent Parameter | Model Equation | Maturity Levels | Regression Constant | | | Statistical Parameters | | The Best Fitted Model |
|---|---|---|---|---|---|---|---|---|---|
| | | | | a | b | c | $R^2$ | *SEE* | |
| $M$ (g) | $W$ (mm) | Linear | Immature | 0.012 | 0.018 | - | 0.794 | 0.003 | Quadratic |
| | | | Mature | 0.062 | 0.012 | - | **0.927** | **0.001** | |
| | | | Ripe | −0.065 | 0.038 | - | 0.807 | 0.020 | |
| $M$ (g) | $W$ (mm) | Quadratic | Immature | 0.306 | −0.110 | 0.014 | 0.895 | 0.003 | |
| | | | Mature | 0.260 | −0.053 | 0.005 | **0.960** | **0.001** | |
| | | | Ripe | 0.557 | −0.186 | 0.020 | 0.926 | 0.014 | |
| $M$ (g) | $W$ (mm) | S-curve | Immature | 0.172 | −0.357 | - | 0.730 | 0.004 | |
| | | | Mature | 0.208 | −0.436 | - | **0.895** | **0.002** | |
| | | | Ripe | 0.346 | −1.094 | - | 0.722 | 0.023 | |
| $M$ (g) | $W$ (mm) | Power | Immature | 0.024 | 0.882 | - | 0.790 | 0.003 | |
| | | | Mature | 0.051 | 0.543 | - | **0.920** | **0.001** | |
| | | | Ripe | 0.010 | 1.523 | - | 0.826 | 0.019 | |
| $M$ (g) | $D_g$ (mm) | Linear | Immature | 0.016 | 0.017 | - | 0.878 | 0.003 | Quadratic |
| | | | Mature | 0.063 | 0.012 | - | **0.945** | **0.001** | |
| | | | Ripe | −0.065 | 0.037 | - | 0.821 | 0.019 | |
| $M$ (g) | $D_g$ (mm) | Quadratic | Immature | 0.225 | −0.071 | 0.009 | 0.938 | 0.002 | |
| | | | Mature | 0.196 | −0.032 | 0.004 | **0.960** | **0.001** | |
| | | | Ripe | 0.479 | −0.152 | 0.016 | 0.886 | 0.017 | |
| $M$ (g) | $D_g$ (mm) | S-curve | Immature | 0.171 | −0.360 | - | 0.822 | 0.003 | |
| | | | Mature | 0.209 | −0.445 | - | **0.922** | **0.001** | |
| | | | Ripe | 0.361 | −1.195 | - | 0.767 | 0.021 | |
| $M$ (g) | $D_g$ (mm) | Power | Immature | 0.026 | 0.840 | - | 0.874 | 0.003 | |
| | | | Mature | 0.051 | 0.541 | - | **0.940** | **0.001** | |
| | | | Ripe | 0.011 | 1.474 | - | 0.832 | 0.018 | |

Data are expressed as $L$, major axis; $T$, medium axis; $W$, minor axis; $D_g$, geometric diameter; $R^2$, coefficient of determination; *SEE*, standard error of estimate; a, b, c, constants of curve fittings.

**Table 3.** Mass models of pepper berries based on volume and surface area.

| Dependent Parameter | Independent Parameter | Model Equation | Maturity Levels | Regression Constant | | | Statistical Parameters | | The Best Fitted Model |
|---|---|---|---|---|---|---|---|---|---|
| | | | | a | b | c | $R^2$ | *SEE* | |
| $M$ (g) | $V$ (mm$^3$) | Linear | Immature | 0.108 | −0.015 | - | 0.636 | 0.005 | Quadratic |
| | | | Mature | 0.049 | 0.001 | - | 0.806 | 0.002 | |
| | | | Ripe | −0.095 | 0.002 | - | **0.816** | **0.027** | |
| $M$ (g) | $V$ (mm$^3$) | Quadratic | Immature | 0.125 | −0.190 | 0.163 | 0.925 | 0.002 | |
| | | | Mature | 0.795 | −0.012 | $5.164 \times 10^{-5}$ | 0.988 | 0.001 | |
| | | | Ripe | 0.828 | −0.015 | $7.376 \times 10^{-5}$ | **0.995** | **0.006** | |
| $M$ (g) | $V$ (mm$^3$) | S-curve | Immature | 0.092 | 0.002 | - | 0.692 | 0.004 | |
| | | | Mature | 0.217 | −9.947 | - | **0.762** | **0.002** | |
| | | | Ripe | 0.375 | −24.784 | - | 0.727 | 0.033 | |
| $M$ (g) | $V$ (mm$^3$) | Power | Immature | 0.096 | $6.619 \times 10^{-18}$ | - | 0.000 | 0.008 | |
| | | | ture | 0.006 | 0.639 | - | 0.798 | 0.002 | |
| | | | Ripe | $2.626 \times 10^{-5}$ | 1.819 | - | **0.849** | **0.024** | |
| $M$ (g) | $SA_{sp}$ (mm$^2$) | Linear | Immature | 0.055 | $6 \times 10^{-4}$ | - | 0.899 | 0.002 | Quadratic |
| | | | Mature | 0.100 | $3 \times 10^{-4}$ | - | 0.952 | 0.001 | |
| | | | Ripe | 0.020 | 0.001 | - | **0.955** | **0.009** | |
| $M$ (g) | $SA_{sp}$ (mm$^2$) | Quadratic | Immature | 0.097 | $-6 \times 10^{-4}$ | $7.825 \times 10^{-6}$ | 0.936 | 0.002 | |
| | | | Mature | 0.124 | $-1 \times 10^{-4}$ | $1.715 \times 10^{-6}$ | 0.960 | 0.001 | |
| | | | Ripe | 0.117 | −0.001 | $7.80 \times 10^{-6}$ | **0.983** | **0.006** | |
| $M$ (g) | $SA_{sp}$ (mm$^2$) | S-curve | Immature | 0.132 | −2.596 | - | 0.789 | 0.003 | |
| | | | Mature | 0.172 | −4.186 | - | **0.905** | **0.002** | |
| | | | Ripe | 0.280 | −12.808 | - | 0.827 | 0.018 | |
| $M$ (g) | $SA_{sp}$ (mm$^2$) | Power | Immature | 0.016 | 0.419 | - | 0.874 | 0.003 | |
| | | | Mature | 0.038 | 0.270 | - | 0.940 | 0.001 | |
| | | | Ripe | 0.002 | 0.885 | - | **0.950** | **0.010** | |

Data are expressed as $V$, actual volume; $SA_{sp}$, spherical surface area; $R^2$, coefficient of determination; *SEE*, standard error of estimate; a, b, c, constants of curve fittings.

**Table 4.** Mass models of pepper berries based on projected area.

| Dependent Parameter | Independent Parameter | Model Equation | Maturity Levels | Regression Constant | | | Statistical Parameters | | The Best Fitted Model |
|---|---|---|---|---|---|---|---|---|---|
| | | | | a | b | c | $R^2$ | SEE | |
| $M$ (g) | $PA_L$ (mm$^2$) | Linear | Immature | 0.055 | 0.002 | - | 0.831 | 0.003 | |
| | | | Mature | 0.098 | 0.001 | - | 0.912 | 0.001 | |
| | | | Ripe | 0.042 | 0.004 | - | 0.792 | 0.020 | |
| $M$ (g) | $PA_L$ (mm$^2$) | Quadratic | Immature | 0.086 | −0.001 | $1 \times 10^{-4}$ | **0.856** | **0.003** | |
| | | | Mature | 0.111 | 0.001 | $1.466 \times 10^{-5}$ | **0.913** | **0.002** | |
| | | | Ripe | 0.158 | −0.005 | $1 \times 10^{-4}$ | **0.835** | **0.020** | Quadratic |
| $M$ (g) | $PA_L$ (mm$^2$) | S-curve | Immature | 0.132 | −0.614 | - | 0.729 | 0.004 | |
| | | | Mature | 0.175 | −1.063 | - | 0.883 | 0.002 | |
| | | | Ripe | 0.252 | −2.626 | - | 0.700 | 0.024 | |
| $M$ (g) | $PA_L$ (mm$^2$) | Power | Immature | 0.029 | 0.422 | - | 0.807 | 0.003 | |
| | | | Mature | 0.054 | 0.280 | - | 0.906 | 0.002 | |
| | | | Ripe | 0.013 | 0.723 | - | 0.782 | 0.021 | |
| $M$ (g) | $PA_T$ (mm$^2$) | Linear | Immature | 0.055 | 0.002 | - | 0.887 | 0.003 | |
| | | | Mature | 0.102 | 0.001 | - | 0.941 | 0.001 | |
| | | | Ripe | 0.040 | 0.004 | - | 0.810 | 0.019 | |
| $M$ (g) | $PA_T$ (mm$^2$) | Quadratic | Immature | 0.114 | −0.004 | $2 \times 10^{-4}$ | **0.946** | **0.002** | |
| | | | Mature | 0.146 | −0.002 | $4.808 \times 10^{-5}$ | **0.971** | **0.001** | |
| | | | Ripe | 0.199 | −0.009 | $2 \times 10^{-4}$ | **0.930** | **0.013** | Quadratic |
| $M$ (g) | $PA_T$ (mm$^2$) | S-curve | Immature | 0.131 | −0.647 | - | 0.783 | 0.004 | |
| | | | Mature | 0.169 | −0.980 | - | 0.874 | 0.001 | |
| | | | Ripe | 0.239 | −2.167 | - | 0.642 | 0.027 | |
| $M$ (g) | $PA_T$ (mm$^2$) | Power | Immature | 0.029 | 0.412 | - | 0.860 | 0.003 | |
| | | | Mature | 0.058 | 0.250 | - | 0.921 | 0.001 | |
| | | | Ripe | 0.013 | 0.741 | - | 0.790 | 0.020 | |

**Table 4.** *Cont.*

| Dependent Parameter | Independent Parameter | Model Equation | Maturity Levels | Regression Constant | | | Statistical Parameters | | The Best Fitted Model |
|---|---|---|---|---|---|---|---|---|---|
| | | | | a | b | c | $R^2$ | SEE | |
| $M$ (g) | $PA_W$ (mm$^2$) | Linear | Immature | 0.052 | 0.003 | - | 0.820 | 0.003 | Quadratic |
| | | | Mature | 0.099 | 0.001 | - | 0.934 | 0.001 | |
| | | | Ripe | 0.039 | 0.004 | - | 0.815 | 0.019 | |
| $M$ (g) | $PA_W$ (mm$^2$) | Quadratic | Immature | 0.112 | −0.005 | $2 \times 10^{-4}$ | **0.888** | **0.003** | |
| | | | Mature | 0.139 | −0.002 | $4.768 \times 10^{-5}$ | **0.955** | **0.001** | |
| | | | Ripe | 0.207 | −0.010 | $3 \times 10^{-4}$ | **0.942** | **0.012** | |
| $M$ (g) | $PA_W$ (mm$^2$) | S-curve | Immature | 0.132 | −0.619 | - | 0.695 | 0.004 | |
| | | | Mature | 0.172 | −1.011 | - | 0.872 | 0.002 | |
| | | | Ripe | 0.240 | −2.168 | - | 0.652 | 0.026 | |
| $M$ (g) | $PA_W$ (mm$^2$) | Power | Immature | 0.027 | 0.441 | - | 0.790 | 0.003 | |
| | | | Mature | 0.055 | 0.271 | - | 0.916 | 0.001 | |
| | | | Ripe | 0.013 | 0.744 | - | 0.795 | 0.020 | |
| $M$ (g) | CPA | Linear | Immature | 0.054 | 0.002 | - | 0.832 | 0.003 | Quadratic |
| | | | Mature | 0.099 | 0.001 | - | 0.950 | 0.001 | |
| | | | Ripe | 0.034 | 0.004 | - | 0.833 | 0.018 | |
| $M$ (g) | CPA | Quadratic | Immature | 0.102 | −0.003 | $2 \times 10^{-4}$ | **0.880** | **0.003** | |
| | | | Mature | 0.130 | −0.001 | $3.55 \times 10^{-5}$ | **0.962** | **0.001** | |
| | | | Ripe | 0.221 | −0.011 | $3 \times 10^{-4}$ | **0.966** | **0.009** | |
| $M$ (g) | CPA | S-curve | Immature | 0.132 | −0.632 | - | 0.716 | 0.004 | |
| | | | Mature | 0.172 | −1.037 | - | 0.896 | 0.002 | |
| | | | Ripe | 0.250 | −2.445 | - | 0.683 | 0.025 | |
| $M$ (g) | CPA | Power | Immature | 0.028 | 0.431 | - | 0.805 | 0.003 | |
| | | | Mature | 0.055 | 0.272 | - | 0.935 | 0.001 | |
| | | | Ripe | 0.011 | 0.784 | - | 0.816 | 0.019 | |

Data are expressed as $PA_L$, projected area perpendicular to major axis; $PA_T$, projected area perpendicular to medium axis; $PA_W$, projected area perpendicular to minor axis; CPA, criteria projected area; $R^2$, coefficient of determination; SEE, standard error of estimate; a, b, c, constants of curve fittings.

### 3.4. Models Based on Volume

In Table 3 which shows the mass prediction results of the pepper berries based on actual volume ($V$), the Quadratic model based on $V$ (Equation (19)) was found to be the best fit when compared to the other models. It had the highest $R^2$ of 0.995 and the lowest *SEE* of 0.006 for ripe pepper berries.

$$M = 0.828 - 0.015\ V + 7.376 \times 10^{-5} V^2 \tag{19}$$

For immature pepper berries, the Quadratic model based on $V$ was suitable with the highest values of $R^2$ and *SEE,* which were 0.925 and 0.002, respectively, as shown in Table 3. Equation (20) was obtained for the pepper berries at the immature maturity level.

$$M = 0.081 - 5.726 \times 10^{-6} V + 3.703 \times 10^{-6} V^2 \tag{20}$$

The mature pepper berries had a Quadratic model based on $V$ as the best model with the highest $R^2$ of 0.988 and the lowest *SEE* of 0.001, shown in Table 3. The Quadratic model equation (Equation (21)) is as follows:

$$M = 0.795 - 0.012\ V + 5.164 \times 10^{-5} V^2 \tag{21}$$

Thus, the ripe pepper berries had the highest value of $R^2$ and lowest *SEE* for actual volume among the maturity levels. Therefore, a quadratic form was shown as the suggested mass model-based volume, similar to the prediction of the Fava bean mass with $R^2 = 0657$ [3].

### 3.5. Models Based on Surface Area

As shown in Table 3 for the results of mass prediction of pepper berries based on surface area ($SA_{sp}$), the Quadratic model was the best based on the highest value of $R^2$ compared to the other models. For the best fit model, the Quadratic model based on $SA_{sp}$ of ripe pepper berries had the highest value of $R^2$ and lowest *SEE* of the surface area-assumed shape; the respective values were 0.984 and 0.006 (as shown in Equation (22)). It was also the best fit among the models of other maturity levels of pepper berries.

$$M = 0.118 - 0.001\ SA_{sp} + 7.900 \times 10^{-6}\ SA_{sp}^2 \tag{22}$$

The Quadratic model was suitable for $SA_{sp}$ which had the highest $R^2$ of 0.936 and lowest *SEE* of 0.002 for immature pepper berries (as shown in Equations (23)). The suitable model of immature pepper berries is shown in the following equation:

$$M = 0.097 - 6 \times 10^{-4}\ SA_{sp} + 7.825 \times 10^{-6}\ SA_{sp}^2 \tag{23}$$

For mature pepper berries, the suitable model equation was Quadratic. Based on Table 3, the parameter $SA_{sp}$ had the highest value of $R^2$ and lowest *SEE,* which were 0.960 and 0.001, respectively. The model equation was formed as indicated in Equation (24).

$$M = 0.124 - 1 \times 10^{-4}\ SA_{sp} + 1.715 \times 10^{-6}\ SA_{sp}^2 \tag{24}$$

Among these maturity levels, the ripe pepper berries had the highest value of $R^2$ and lowest *SEE* for the spheroid surface area in Quadratic form. Therefore, the Quadratic form was shown as the suggested mass model.

*3.6. Models Based on Projected Area*

Among the models based on the projected area ($PA_L$, $PA_T$, $PA_W$, and $CPA$), the Quadratic model comprising $PA_T$ was the best fit with the highest $R^2$ of 0.71 and lowest *SEE* of 0.001 for mature pepper berries as shown in Table 4. Equation (25) shows the model equation obtained. The Quadratic model based on $PA_T$ (Equation (26)) was suitable, with $R^2$ of 0.946 and *SEE* of 0.002, with respect to immature pepper berries. The Quadratic model based on $PA_W$ (Equation (27)) also achieved a suitable $R^2$ of 0.942 with *SEE* of 0.012 for ripe pepper berries.

$$M = 0.123 - 0.002PA_T + 4.808 \times 10^{-5}PA_T{}^2 \tag{25}$$

$$M = 0.114 - 0.004PA_T + 2 \times 10^{-4}PA_T{}^2 \tag{26}$$

$$M = 0.207 - 0.010PA_W + 3 \times 10^{-4}PA_W{}^2 \tag{27}$$

Furthermore, the Quadratic model of the projected area based on the criteria projected area (*CPA*) shown in Table 4 was preferred as the best model to calculate the mass of pepper berries. Due to the high $R^2$ value of 0.966, a model based on *CPA* (Equation (28)) for ripe pepper berries was found as the best fit with *SEE* of 0.009. The mature pepper berries had values of $R^2$ and *SEE* (Equation (29)) which were 0.962 and 0.001, respectively. The immature pepper berries had the lowest $R^2$ and *SEE* values for the Quadratic model based on *CPA* (Equation (30)), which were 0.880 and 0.003, respectively, compared to other maturity levels of pepper berries.

$$M = 0.221 - 0.011CPA + 3 \times 10^{-4}CPA^2 \tag{28}$$

$$M = 0.130 - 0.001CPA + 3.55 \times 10^{-5}CPA^2 \tag{29}$$

$$M = 0.102 - 0.003CPA + 2 \times 10^{-4}CPA^2 \tag{30}$$

Thus, all three projected areas are necessary to be specified and applied in grading the pepper berries by using this model.

**4. Conclusions**

In the current study, the mass of ripe pepper berries based on volume (Quadratic model) is the recommended equation, as the nonlinear form $M = 0.828 - 0.015V + 7.376 \times 10^{-5}V^2$ had the highest $R^2$, 0.995, and lowest *SEE*, 0.006, when compared to other maturity levels of pepper berries. This shows a very good relationship between the mass and actual volume of pepper berries. The model predicting the mass of pepper berries considered as spheroid was found to be the most applicable (Quadratic model is recommended). Finally, the Quadratic model is applicable to all properties due to its economical viewpoint. The mass model of pepper berries based on actual volume in the obtained results is recommended for designing and optimizing machines for handling, cleaning, conveying, and storing.

**Author Contributions:** P.N.M.A.A. conducted the experiments, collected and analyzed the data of results, and wrote the manuscript. R.S. supervised the research and revised the manuscript. H.C.M. and M.E.Y. supervised the experiments. All authors have read and agreed to the published version of the manuscript.

**Funding:** This research received no external funding.

**Acknowledgments:** The authors would like to express their appreciation to Universiti Putra Malaysia for the financial and technical support given during this research work.

**Conflicts of Interest:** The authors declare no conflict of interest.

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
