# Peer review of "Some Physical Properties and Mass Modelling of Pepper Berries (Piper nigrum L.), Variety Kuching, at Different Maturity Levels"

_processes, doi:10.3390/pr8101314_

Round 1
Reviewer 1 Report
The authors should try to present the data in a more friendly way to the readers. The discussion is very weak.
1. General the problem is that the article not show why these parameters was measured except for one sentence in the conclusions line 379: "The mass model of pepper berries based on actual volume in the obtained results is recommended for designing and optimizing the machine for handling, cleaning, conveying, and storing." If its enough?
2. For me the topic is not so interesting, however the modeling is difficult to present interesting.
3. In the paper the results are presented just in the tables and still missing the discussion, which are not clearly presented and difficult to read. So the results should be clearly presented and presented in friendly way. The conclusions are correct.
My comments to the article are listed directly in the pdf file.

Reviewer 2 Report
Study present the influences of some physical properties on the mass of one type of pepper and one correlation is presented after an experimental and statistical study. Method used to have the volume of the product can be improved using a liquid less densified than the product, with applied the displacement of this liquid. Thus, it is possible to have a representative error to have each volume per sample. However, because it is importnt to predict the mass of products during modelling of heat mass transfer, I suggest that the paper is interesting. But some corrections are capital before to be accepted to be published.
- Please, delete the first and second sentences of the conclusion section. These two sentences not have any relation with the title. Please, the title and abstract don’t give the names of physical properties you studied. Can you give more precisions on physical properties you studied in abstract section ?
- Keywords : change pepper (common name) by the scientific name used in the title. Modelling is a prediction, delecte ‘’prediction’’. Change ‘’Mass modelling’’ by ‘’Modelling’’. Change ‘’Physical properties’’ by the names of all properties you studied.
- Introduction section : page 2 lines 80 and 81 : precise the properties you studied.
- Page 3 line 94 : give more explanations on the method to select the good quality of the product. Also, gie the age to have immature, mature and Ripe products.
- Page 3 lines 98-100 : gives a reference that confirms the number of samples to have a representative results of physical properties.
- Lines 106-110 : I don’t understand : you used an assumption that the samples are spheric or ellipsoid !! I don’t agree this assumption. A representative error is doing in each sample, and on 100 samples, error will be important. The method of liquid displadement is the good. It was very interessing to select a liquid less densified than the studied product. However, can you cite more papers (3 or four) used your assumption ? also, precise the errors after using this assumption.
- Page 18 : please delete the first paragraph because, I believe that I read it in introduction section. In effect, it place is in introduction section or/and in Method section.
- Lines 263-369. I dont agree this method to use X parameter. I suggest authors to use each reference of physical parameters. For example, when X is Dg, present M in relation to Dg and not to X each time. Use X only when you want to generalise.
Round 2
Reviewer 1 Report
The authors correct the manuscript according to reviewer comments.
Author Response
We are very thankful for the reviewers' efforts in evaluating the manuscript and providing constructive comments and substantial suggestions in improving the manuscript quality.
Reviewer 2 Report
Authors not give an answer to all my questions, but the manuscript can be accepted in this form.
Author Response

(The authors gave the same response as above.)
